# Rapid Visual Detection of *Peronophythora litchii* on Lychees Using Recombinase Polymerase Amplification Combined with Lateral Flow Assay Based on the Unique Target Gene *Pl_101565*

**DOI:** 10.3390/plants13040555

**Published:** 2024-02-18

**Authors:** Rongbo Wang, Benjin Li, Mingyue Shi, Yumei Zhao, Jinlong Lin, Qinghe Chen, Peiqing Liu

**Affiliations:** 1Fujian Key Laboratory for Monitoring and Integrated Management of Crop Pests, Institute of Plant Protection, Fujian Academy of Agricultural Sciences, Fuzhou 350003, China; wangrongbo@faas.cn (R.W.); libj@faas.cn (B.L.); shi17750253482@126.com (M.S.); 18328818578@163.com (Y.Z.); lqrlwlch@126.com (J.L.); 2Sanya Institute of Breeding and Multiplication, School of Tropical Agriculture and Forestry, Hainan University, Sanya 572000, China; qhchen@hainanu.edu.cn

**Keywords:** *Peronophythora litchii*, recombinase polymerase amplification, lateral flow, lychee

## Abstract

Downy blight, caused by *Peronophythora litchii*, is a destructive disease that impacts lychee fruit throughout the pre-harvest, post-harvest, and transportation phases. Therefore, the prompt and precise identification of *P. litchii* is crucial for the effective management of the disease. A novel gene encoding a Rh-type ammonium transporter, *Pl_101565*, was identified in *P. litchii* through bioinformatic analysis in this study. Based on this gene, a coupled recombinase polymerase amplification–lateral flow (RPA-LF) assay for the rapid visual detection of *P. litchii* was developed. The assay has been shown to detect *P. litchii* accurately, without cross-reactivity to related pathogenic oomycetes or fungi. Moreover, it can be performed effectively within 15 to 25 min at temperatures ranging from 28 to 46 °C. Under optimized conditions, the RPA-LF assay could detect as low as 1 pg of *P. litchii* genomic DNA in a 25 μL reaction system. Furthermore, the RPA-LF assay successfully detected *P. litchii* in infected lychee samples within a 30 min timeframe. These attributes establish the RPA-LF assay as a rapid, sensitive, and specific method for diagnosing *P. litchii* early; it is particularly suitable for applications in resource-limited settings.

## 1. Introduction

Lychees (*Litchi chinensis* Sonn.), which are a globally cultivated fruit crop of substantial commercial importance, are prevalent in subtropical and tropical regions [1,2]. In China, their cultivation is predominantly concentrated in the southern provinces, encompassing Fujian, Guangxi, Guangdong, and Hainan [3]. Among the foremost challenges confronting the lychee industry in these regions, the incidence of lychee downy blight is noticeable, which is instigated by the oomycete pathogen *Peronophythora litchii* and influences all lychee-growing areas in China [4,5]. It significantly impacts lychee production, contributing to an annual loss of about 20–30% of the fruit yield [6]. This disease is one of the most serious threats to lychee crops throughout the pre-harvest, post-harvest, and transportation phases, leading to the rotting of tender leaves, flowers, and fruits [7]. The latent infection of harvested lychee fruit by *P. litchi* is particularly concerning, which often leads to significant damage to the fruit and a significantly shortened shelf life [8]. This latent infection is a primary factor contributing to the rapid deterioration in the quality of harvested lychees [6,7]. Therefore, the prompt and precise identification of *P. litchii* is crucial for effectively managing the disease.

The conventional diagnostic methodology for plant pathogens, which encompasses the sequential steps of isolation, culture, and pathogenicity testing in accordance with Koch’s postulates, is a time-consuming process. Currently, molecular diagnostic approaches based on nucleic acids have become the preferred method for pathogen detection [9]. The conventional polymerase chain reaction (PCR) has been the most frequently utilized among these methods [10,11]. However, the applicability of conventional PCR methods in field settings is limited due to their lengthy timeframes and the need for specialized equipment. Recently, Kong et al. (2021) [12] developed a loop-mediated isothermal amplification (LAMP) assay for detecting *P. litchii*, which was incubated at 60–65 °C for approximately 60 min. However, this method requires the design of four specific primers for six distinct regions on the target gene [13]. Accordingly, the requirements for the primer design are stringent, and designing primers that can differentiate closely related species is often challenging.

Recombinase polymerase amplification (RPA) is a remarkable isothermal DNA amplification technology that is widely used in pathogen detection. This innovative method rapidly amplifies the target nucleic acids directly from samples, requiring minimal processing [14]. One of the notable features of RPA is its use of portable instruments, making it highly accessible for various settings including field applications [15]. A key advantage of RPA over the LAMP assay lies in its operational temperature range. RPA functions effectively at lower temperatures, typically those between 37 to 42 °C, which are less stringent than LAMP’s temperature requirements [16]. RPA offers enhanced specificity in targeting nucleic acid sequences, requiring only two primers for the sequence-specific recognition of the target sites. Additionally, the RPA process is notably fast. Typically, detectable amplification products are obtained within 20 min [17]. Furthermore, incorporating an oligonucleotide probe into the RPA assay enables the visualization of amplification products through a lateral flow dipstick (LF). This method is portable, accurate, and rapid, thereby making it suitable for resource-limited settings [18]. Given its merits, the RPA-LF assay has been applied extensively for the detection of diverse pathogens, including bacteria [19,20], fungi [21], parasites [18], viruses [22], and more. Recently, numerous RPA-LF assays have been successfully developed for detecting oomycete pathogens, such as *Phytophthora sojae* [23], *Ph. infestans* [24], *Ph. capsici* [25], *Ph. parasitica* [26], and *Pythium* [27].

In this investigation, the fusion of RPA and LF methodologies was employed to propose a rapid detection assay for *P. litchii*, which is based on the newly identified gene *Pl_101565* encoding a Rh-type ammonium transporter that has been derived from genomic sequence data. The specificity of the RPA-LF assay was assessed through testing against various oomycete and fungal species. Moreover, an attempt was made to compare the sensitivity of the RPA-LF assay with that of a conventional PCR-based approach. The developed assay was analyzed using samples artificially inoculated with *P. litchii* to assess practical applicability.

## 2. Results

### 2.1. Bioinformatical Characterizations of the Specific Target Gene Pl_101565 from P. litchii

The family of ammonium transport proteins encompasses a diverse range of proteins, including the methylammonium permeases (Mep) from yeast, the ammonium transport proteins (Amt) found in plants, archaea, and bacteria, and the Rhesus (Rh) proteins in animals [28,29]. In this study, a unique Rh-type ammonium transporter gene *Pl_101565*, was identified in *P. litchii*, which was conserved among oomycetes, exhibiting sequence similarity ranging from 44.93% to 79.87%. Notably, no homologs of the *Pl_101565* gene were detected in either plants or fungi (Figure 1). Therefore, due to its conservation in oomycetes along with sufficient variability in certain sites, the *Pl_101565* gene emerged as an ideal target for diagnostic assays.

### 2.2. Establishment of the RPA-LF Assay for P. litchii Based on the Pl_101565 Gene

Primer pairs and a probe for the RPA assay were formulated by analyzing specific segments of the *Pl_101565* gene, drawing comparisons among the Rh sequences obtained from *P. litchii* and closely related oomycete species (Figure 2). Subsequently, the RPA-LF assay was performed using the *P. litchii* genomic DNA as a template to assess the efficacy of the specific primer pairs and the probe. As shown in Figure 3A, the dipsticks used in the assay displayed a visible control line, confirming the validity of the tests. In the positive reaction using the gDNA of *P. litchii*, the dipstick showed both the visible test and control lines. Conversely, only the control line was observed in the negative assay. Additionally, an approximately 194 bp long RPA amplicon was detected in the positive reaction containing the gDNA of *P. litchii*, but it was absent in the negative control (Figure 3B). These findings suggest that the designed primer pairs and probe are effective in visually detecting *P. litchii* using the RPA-LF assay.

### 2.3. Optimal Conditions for the RPA-LF Assay

The reaction temperature and duration were evaluated using 100 ng of *P. litchii* gDNA as the template to establish the optimal reaction conditions. The reaction mixtures were incubated over temperatures ranging from 25 to 46 °C with intervals of 3 °C for 30 min. The results indicated that the test lines on the LF strips were visible across temperatures ranging from 28 to 46 °C. Notably, no significant increase was observed in the density of the test line from 37 to 46 °C (Figure 4A). Therefore, a reaction temperature of 37 °C was identified as the optimal condition. Additionally, the amplification time of RPA was assessed to be 37 °C for various amplification durations (5, 10, 15, 20, 25, 30, and 35 min). The test line was faintly observable after 15 min of reaction time, which was followed by the emergence of a clearly positive test band evident after 25 min (Figure 4B). Consequently, a 25 min reaction time was discerned as the optimal one for the RPA-LF assay of *P. litchii*, particularly in the context of rapid diagnostics. The experiments were replicated three times with consistent results.

### 2.4. Specificity of the RPA-LF Assay

Seven *P. litchii* isolates, nine closely related oomycete isolates, and seven fungi were tested under the abovementioned conditions to assess the specificity (Table 1). The results indicated visible test lines on dipsticks for all seven *P. litchii* isolates from various regions of China, while dipsticks for other pathogenic species and NTCs exhibited no test lines (Figure 5). The consistent results of the experiment were obtained in all three repetitions. Therefore, the designed primer pair–probe combination demonstrated itself to be highly specific for *P. litchii*, with no observed cross-reactivity with other plant pathogenic species.

### 2.5. Detection Sensitivity of the RPA-LF Assay

To assess sensitivity, a set of 10-fold serial dilutions of *P. litchii* gDNA ranging from 10 ng/μL to 10 fg/μL were used as templates for detection. As shown in Figure 6, the emergence of a faint test band at 1 pg/μL of DNA revealed that the RPA-LF assay can detect *P. litchii* genomic DNA at a minimum concentration of 1 pg in a 25 μL reaction volume. The results for the sensitivity assay were consistent across all three replicates.

### 2.6. Detection of P. litchii in Artificially Inoculated Lychee Leaves and Fruits Using the RPA-LF Assay

To confirm the reliability of the RPA-LF method, we analyzed different lychee samples that were artificially inoculated with *P. litchii*. Five days post-inoculation, the infected lychee leaves were divided into four distinct regions, with region 1 being the furthest from the site of inoculation (Figure 7A). Similarly, lychee fruits inoculated with *P. litchii* displayed characteristic symptoms of downy blight in contrast with the non-inoculated control fruits (Figure 7B).

In the RPA-LF assay, all the lateral flow strips exhibited a clearly visible control line. Test bands were present on the dipsticks used for testing genomic DNA extracted from regions 2, 3, and 4 of the inoculated leaf as well as from the lychee fruits rather than in region 1 or the non-inoculated samples (Figure 7C). Additionally, all the artificially inoculated samples underwent testing through a conventional PCR assay. The results showed that the amplification bands were observed in PCR reactions containing genomic DNA from the inoculated lychee leaf (regions 3 and 4) and fruit samples (Figure 7D). However, no bands were observed in the sample from region 2, which could be attributable to the lower sensitivity of the PCR assay relative to the RPA-LF method. Three replications of the experiments produced identical results. Therefore, the developed RPA-LF assay could be suitable for detecting *P. litchii* in its natural host environment.

## 3. Discussion

Lychees (*Litchi chinensis* Sonn.) are a fruit of significant commercial importance in subtropical countries, and they often face challenges during post-harvest storage. The ripe fruit is susceptible to browning and rotting, primarily due to various diseases, thereby resulting in a substantially reduced shelf life. One of the major diseases impacting *litchi* is that of downy blight, leading to annual commercial losses of approximately 20–30% [1,3,7,8]. Consequently, the prompt and precise detection of *P. litchii* is vital for the timely management of the disease and for preventing the costs associated with a misdiagnosis. A previous study developed a LAMP assay for specifically detecting *P. litchii*, which operates at 65 °C for 60 min [12]. However, the RPA-LF assay offers advantages in terms of speed and operational conditions, with an amplification time that is only 30 min and a lower incubation temperature of 39℃. Notably, this represents the inaugural report utilizing the RPA-LF assay for the rapid visual detection of *P. litchii*.

The selection of appropriate molecular markers is crucial for enhancing the specificity and sensitivity of pathogen detection. Various gene fragments, including the ras-related protein gene *Ypt1*, the internal transcribed spacer (*ITS*) region, and mitochondrial genes such as *Cox 1* and *Cox 2*, have been employed for the molecular identification of oomycete pathogens [9,23,30,31]. Nevertheless, the limited sequence variability among closely related species often constrains their effectiveness in distinguishing specific pathogens. With the increasing prevalence of pathogen genomic sequencing, bioinformatic tools are progressively employed to pinpoint targets for molecular diagnostics. The utilization of DNA fragments as molecular markers for the detection of oomycetes is on the rise [9].

A comparative genomic approach has led to the successful identification of unique target genes. For instance, a spore wall protein gene was identified and used to detect *Pythium ultimum* [32]. Similarly, a novel effector gene, *PHYCI_587572*, was developed as a specific biomarker for detecting *P. cinnamomi* using the RPA-LFD assay [33]. In this study, the Rh-type ammonium transporter gene, *Pl_101565*, was identified from the *P. litchii* genomic sequence. No homologs of gene *Pl_101565* were identified in other reference plant and fungal species, suggesting that this gene is specific to the oomycete species. Additionally, a comparative analysis of *P. litchii* and nine other oomycetes revealed sequence similarities between 44.93% and 79.87%, indicating that gene *Pl_101565* exhibited sufficient sequence variability among different oomycetes. Furthermore, the specificity evaluation of the novel RPA-LF assay showed that it successfully detected the DNAs of *P. litchii* without cross-reacting with those of the 16 other oomycete and fungal species. Nevertheless, to further confirm the inclusivity and reliability of this marker, additional testing with more isolates is recommended in future studies.

Sensitivity is a critical aspect of molecular detection technologies, with higher sensitivity leading to the more effective detection of the target pathogens in samples. The RPA-LF assay exhibited a detection limit of 1 pg in a 25 μL RPA reaction in this study. Under the same reaction volume, it is significantly higher than that of the previously used LAMP method for detecting *P. litchii*, being at least 10 times more sensitive. Moreover, it is 1000 times more sensitive compared with that of a conventional PCR assay employing M90F and M90R primers [12]. Additionally, the sensitivity reported in this study surpassed that of previous reports for detecting gDNA amounts in various *Phytophthora* species, including *Ph. sojae* [23], *Ph. capsici* [25], *Ph. cambivora* [34], *Ph. cinnamomi* [33,35], and *Ph. hibernalis* [36]. This enhanced sensitivity underscores its potential as a highly effective tool for pathogen detection in molecular diagnostics.

The RPA-LF assay developed in this study offers several advantages, particularly under time- and resource-constrained conditions. This study demonstrated that RPA reactions could be performed effectively at 28 to 46 °C. This feature eliminates the need for specialized equipment such as real-time PCR instruments or PCR thermal cyclers. Additionally, the RPA assays described here require relatively short incubation times, typically from 15 to 25 min, with less than 5 min needed for LF detection. In comparison, LAMP assays normally take twice as long to complete, and a typical 30-cycle PCR procedure requires at least 90 min [11,12,32]. However, there are some limitations to RPA assays [18]. One major limitation to consider is that the RPA kits are currently only sold by one company, which may have an impact on pricing. Additionally, there is currently no software available for designing primers specifically for RPA, which can be time-consuming for primer sequence optimization.

Moreover, the results can be rapidly visualized on lateral flow assays that use probes. This method has been effectively applied for detecting various plant pathogens [16,17,18]. Total DNAs encompassing the host and pathogen gDNAs were extracted from lychee plant tissues inoculated with *P. litchii*. Notably, the assays yielded no false-negative results. Therefore, the established RPA-LF assay represents a promising approach for detecting *P. litchii* in field-based molecular diagnosis, offering a practical solution for rapid and accurate pathogen identification.

## 4. Materials and Methods

### 4.1. Pathogen Cultivation and DNA Extraction

A total of 25 isolates of oomycete and fungal species were used in this study, including 7 isolates of *P. litchii*, 9 of other oomycete species, and 9 of different fungal species, as detailed in Table 1. These isolates were maintained at the Institute of Plant Protection, Fujian Academy of Agricultural Sciences, China. Isolates of *P. litchii* and other oomycete species were cultured on a 10% vegetable (V8) juice agar medium at temperatures within the range of 18 to 25 °C in the dark. Pure cultures of the different fungal isolates were cultivated in the dark at 25 °C in a potato dextrose agar medium.

For DNA extraction, mycelia obtained from each isolate were cultured in a 10% liquid V8 medium and a potato dextrose broth medium, respectively, at temperatures ranging from 18 to 25 °C for 3–5 days until the mycelium covered the Petri dish. Subsequently, these mycelia were harvested through filtration and preserved by freezing at −20 °C. The genomic DNA (gDNA) extraction from these cultures was conducted utilizing the FastPure Plant DNA Isolation Mini Kit (Vazyme, Nanjing, China). This was attempted to determine the concentration of DNA in each isolate, which was quantified using a NanoDrop 2000 c spectrophotometer (Thermo Fisher Scientific, Wilmington, DE, USA). Following quantification, the DNA samples were appropriately diluted to the required concentration and stored at −20 °C until they were requisitioned for further analysis.

### 4.2. Identification of the Specific Target Gene for P. litchii

The published genome sequences of *P. litchii* (Accession No. GCA_002812785.1) and nine other sequenced pathogens, including *Ph. sojae* (GCA_000149755.2), *Ph. capsici* (GCA_030324255.1), *Ph. ramorum* (GCA_020800215.1), *Ph. parasitica* (GCA_000247585.2), *Ph. cinnamomi* (GCA_018691715.1), *Ph. infestans* (GCA_000142945.1), *Ph. palmivora* (GCA_008079305.1), *Hyaloperonospora parasitica* (GCA_029452305.1), *Pythium aphanidermatum* (GCA_000387445.2), and *Colletotrichum gloeosporioides* (GCF_011800055.1), were retrieved from the National Center for Biotechnology Information (NCBI) database with the genome coverage ranging from 8.0 to 340.0X [4,32]. All the gene sequences from *P. litchii* were used as queries to perform homology searches against the genomic sequence of *Colletotrichum gloeosporioides* by using BLAST with an E-value cutoff of 1 × 10^−5^. Subsequently, all the retrieved genes were identified and confirmed in the NCBI database. Among the unique identified genes, *Pl_101565* was selected as the candidate diagnostic target as a single-copy gene. The functional analysis of *Pl_101565* was identified by conducting a BLAST search in the InterPro database (https://www.ebi.ac.uk/interpro/, accessed on 15 September 2023) [37].

### 4.3. Primers and Probe Design

To identify Rh-type ammonium transporter genes, HMMER searches based on pfam00909 were performed within the genomes of nine other oomycete species using a cutoff of 1 × 10^−5^ [28]. The candidate sequences were then confirmed iteratively by comparison with the NCBI database. Multiplex sequence alignment analysis, using Clustal W [38], was conducted to identify the specific regions of the gene *Pl_101565* in *P. litchii*. RPA primers and a probe were meticulously designed adhering to the guidelines provided with the TwistAmp^®^ DNA Amplification Kit Assay Design Manual (TwistDx Ltd., Cambridge, UK). The forward and reverse primer lengths of RPA primers were both 30 nt, and the designed probe was 46 nt long. To visualize the results using LF detection, biotin was labeled with the reverse primer PlRPALF-R at the 5′ end. In addition, a specific internal probe, PlRPALF-P, was engineered with a 5′ fluorescein (FAM) label, a 3′ modification with C3 spacer (SpC3), and a base analog tetrahydrofuran (THF) replacing the 31st nucleotide (Table 2). These primers and the probe were synthesized by Sangon Biotech (Shanghai, China).

### 4.4. RPA-LF Assay

The assay was performed following the guidelines outlined in the TwistAmp nfo kit (TwistDx Ltd., Cambridge, UK). Briefly, each RPA reaction (50 μL) comprised rehydration buffer (29.5 μL), primers (2.1 μL of each primer; PlRPALF-F and PlRPALF-R at 10 μM), probe PlRPALF-P (0.6 μL; 10 μM), nuclease-free water (nfH_2_O; 12.2 μL), and the DNA template (1 μL; 100 ng/μL). The reaction mixtures were thoroughly mixed using a vortex, and then each mixture was transferred to the corresponding freeze-dried reaction pellet and mixed by pipetting. To initiate amplification, 2.5 μL of 280 mm magnesium acetate included in the kit was added to each reaction. The RPA reactions were performed at 37 °C for 30 min in a T100 thermal cycler (BIO-RAD, Hercules, CA, USA) with a brief vortexing after the first 5 min. After amplification, each reaction’s RPA product (5 μL) was combined with the HybriDetect assay buffer (100 μL; Milenia Biotec GmbH, Giessen, Germany). Subsequently, the LF strip’s sample pad was immersed in the mixed solution for up to 5 min at room temperature until a distinctly visible control line emerged. Positive detection was confirmed when both the test and control lines appeared concurrently, while the presence of solely the control line signified a negative result. Experiments were performed at least in triplicate. All strips were then air-dried and then documented through photography using a camera.

### 4.5. Optimization of the RPA-LF Conditions

Typically, the RPA reaction can be performed within 20 min at temperatures between 22 and 45 °C [16]. To determine the optimal temperature for the established RPA-LF assay, eight different incubation temperatures (25, 28, 31, 34, 37, 40, 43, and 46 °C) were evaluated, and 100 ng of *P. litchii* genomic DNA were used as a template. To determine the optimal detection time, RPA reactions were carried out at 37 °C for times of 5, 10, 15, 20, 25, 30, and 35 min, with each assay using 100 ng of genomic DNA as a template. The amplicons were promptly subjected to LF strip detection. All tests were performed in triplicate under identical conditions.

### 4.6. Specificity of the RPA–LF Assay

The specificity was assessed using a collection of isolates comprising seven from *P. litchii*, nine oomycete isolates, and seven fungal isolates, as detailed in Table 1. In each reaction, 1 μL of DNA template (100 ng/μL) was used. nfH_2_O served as the negative control in each set of reactions. The RPA process was conducted at 37 °C for 30 min. The assay was repeated three times for each isolate to ensure reliability, maintaining identical conditions for all repetitions.

### 4.7. Sensitivity of the RPA–LF and PCR Assays

To evaluate the sensitivity, a series of ten-fold serial dilutions of *P. litchii* genomic DNA ranging from 10 ng to 10 fg were utilized as templates in the RPA-LF assay. Nuclease-free water (nfH_2_O) served as the negative control in each set of reactions. The RPA-LF reactions were conducted at 37 °C for 30 min, followed by the analysis of amplification products through LF strips. In order to ensure the validity and consistency of the results, all tests were performed in triplicate under identical conditions.

### 4.8. Detection of P. litchii in Artificially Inoculated Lychee Leaves and Fruits Using the RPA-LF Assay

The practical applicability of the developed RPA-LF assay was demonstrated by detecting lychee tissues that were artificially inoculated with *P. litchii*. Lychee leaves were collected from Fujian Agriculture and Forestry University, and the fruits were purchased from a supermarket in Fuzhou, China. The procedure commenced with a thorough cleansing of the lychee leaves and fruits using distilled water, followed by a brief immersion in 75% ethanol for 10 s and subsequent rinsing with distilled water. For inoculation, a 5 mm mycelial plug taken from a fresh *P. litchii* culture was placed on the lesion sites of three replicates of both the leaves and fruits [39]. As a negative control, a 5 mm V8 agar plug without mycelia was used for inoculation. Following inoculation, all samples were placed on filter papers moistened with sterilized water to maintain adequate humidity and incubated in the dark at 25 °C for 5 days. Leaf samples from the inoculated site were segmented into four sections (1, 2, 3, and 4). Total DNA was extracted from 0.1 g of the artificially inoculated plant materials and then adjusted to a concentration of 100 ng/μL for the RPA-LF assay. Purified genomic DNA of *P. litchii* and nfH_2_O was used as the positive and no-template controls (NTCs), respectively. These experiments were conducted three times to ensure consistency.

Additionally, to confirm the results of the RPA-LF assay, conventional PCR was also performed to analyze the genomic DNA from the samples using Yph1F/Yph2R primers [40]. The PCR amplification was carried out within a 25 μL reaction mixture, consisting of 1 μL of each diluted DNA sample, 0.5 μM of both reverse and forward primers, and 12.5 μL of 2× Taq Master Mix (Vazyme, Nanjing, China). The PCR protocol encompassed an initial 3 min denaturation phase at 95 °C, followed by 30 cycles (95 °C for 15 s, 55 °C for 30 s, and 72 °C for 30 s), with a final 10 min extension at 72 °C. PCR products (5 μL) were analyzed by electrophoresis on 1% (wt/vol) agarose gels stained with ethidium bromide (0.5 μg/mL). Similarly to the RPA-LF assay, the PCR assay was thrice replicated to ensure the reproducibility and reliability of the results.

## 5. Conclusions

In conclusion, a new target gene, *Pl_101565*, was bioinformatically identified in this study from the genomic sequence of *P. litchii*. Based on the new biomarker, a novel RPA-LF method was developed for the prompt, precise, and visual identification of *P. litchii*. This versatile assay operates efficiently at 28 to 46 °C and completes the detection process in less than 30 min. The sensitivity evaluation revealed its ability to detect as little as 1 pg of *P. litchii* gDNA in a 25 μL reaction system. Additionally, the assay effectively detected *P. litchii* in artificially inoculated plant tissue samples. Given these results, the RPA-LF assay demonstrated significant potential for application as a routine test for detecting *P. litchii*, especially in field settings where resources are limited. This innovative method offered a fast and convenient molecular tool for the early detection of *P. litchii*, which could facilitate the timely development of proper management strategies for downy blight control, which are particularly important for post-harvest litchi fruit.

## Figures and Tables

**Figure 1 plants-13-00555-f001:**
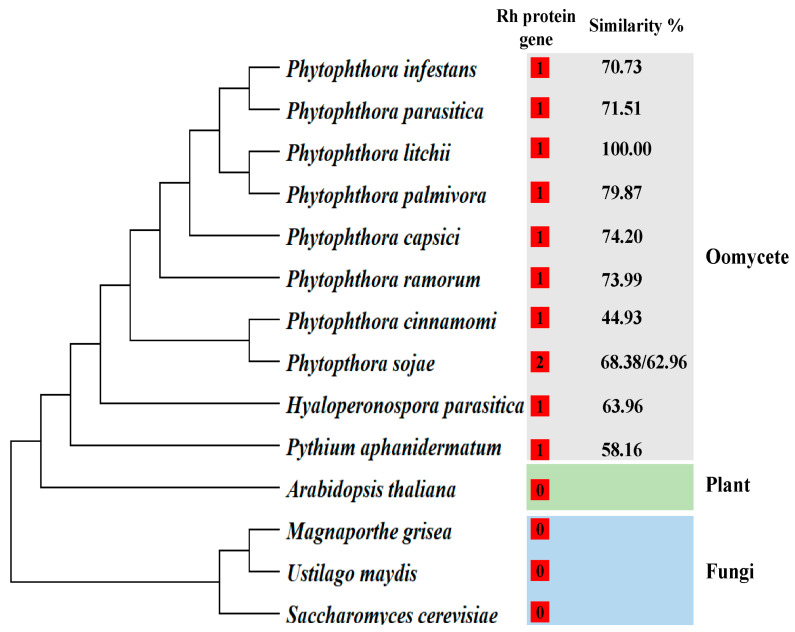
Distribution and similarity of homologs of the *Pl_101565* gene in different species. The numbers in red squares represent the number of homologous genes of *Pl_101565*.

**Figure 2 plants-13-00555-f002:**
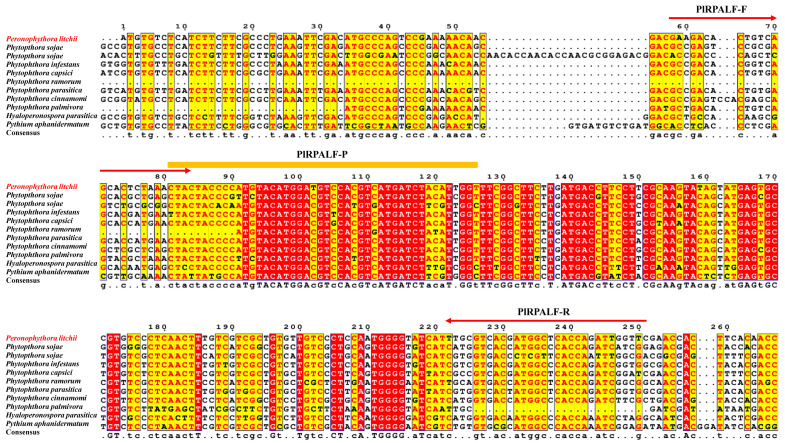
Design of the RPA-LF primers and probe specific for *Peronophythora litchii* based on the newly identified target gene *Pl_101565*. The multiple sequence alignment of *Pl_101565* and its homologs among *P. litchii* and nine related oomycetes and the arrangement of RPA-LF primers and probe are shown. The diagram illustrates the sequences of the RPA-LF forward and reverse primers with red arrows as well as the targeted nucleotides through the probe with a yellow line. Arrows indicate the direction of the extension.

**Figure 3 plants-13-00555-f003:**
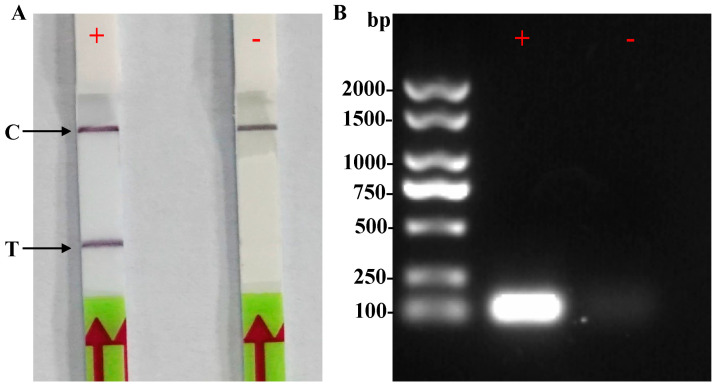
Development of the RPA-LF assay for *Peronophythora litchii*. (**A**) Visual inspection of RPA products utilizing the lateral flow dipstick (LF). (**B**) Detection of RPA products through agarose gel electrophoresis. +: purified genomic DNA of *P. litchii*; -: nuclease-free water (nfH_2_O) as a negative control; M: DL 2000 DNA marker; C: control line; T: test line.

**Figure 4 plants-13-00555-f004:**
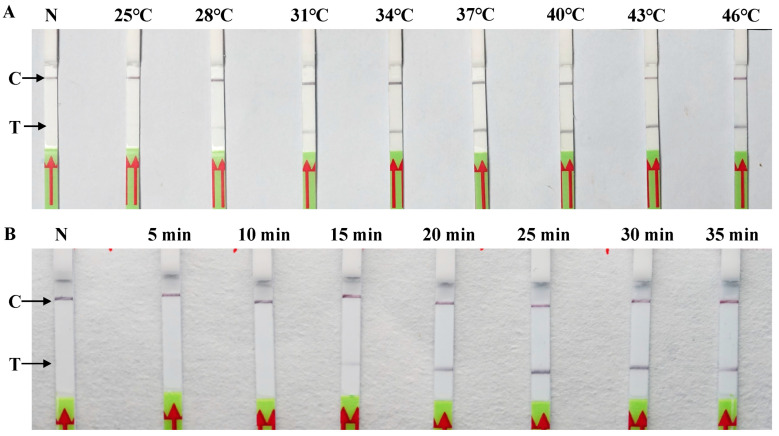
Optimizing the RPA-LF assay. (**A**) Optimization of the RPA amplification temperature. (**B**) Optimization of the RPA amplification time. The top section of each image displays varying temperatures and durations. C: control line, T: test line, N: negative control (nfH_2_O). Three repeats of each evaluation were conducted.

**Figure 5 plants-13-00555-f005:**
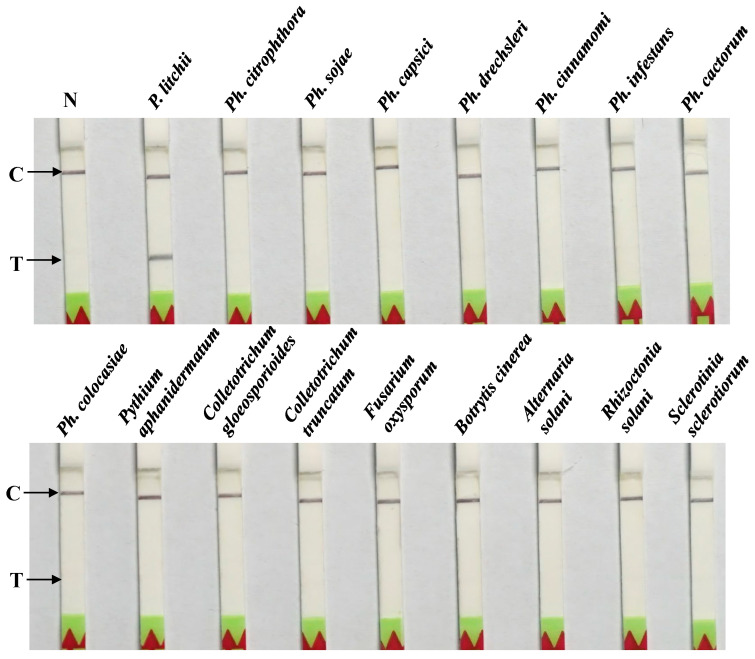
Assessing the specificity of the RPA-LF assay. N: negative control (nfH_2_O); 1: *Peronophythora litchii*; 2: *Phytophthora citrophthora*; 3: *P. sojae*; 4: *P. capsici*; 5: *P. drechsleri*; 6: *P. cinnamomi*; 7: *P. infestans*; 8: *P. cactorum*; 9: *P. colocasiae*; 10: *Pythium aphanidermatum*; 11: *Colletotrichum gloeosporioides*; 12: *Colletotrichum truncatum*; 13: *Fusarium oxysporum*; 14: *Botrytis cinerea*; 15: *Alternaria solani*; 16: *Rhizoctonia solani*; 17: *Sclerotinia sclerotiorum*. C: control line and T: test line. The results of the experiment were identical in all three repetitions.

**Figure 6 plants-13-00555-f006:**
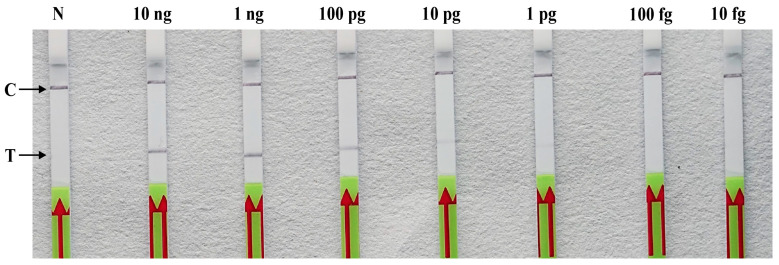
Detection sensitivity of the RPA-LF assay and conventional PCR. Serial dilutions of *P. litchii* gDNA were utilized as the templates for the RPA-LF and PCR assays to evaluate the detection limit. N: negative control (nfH_2_O), C: control line, T: test line. The experiment was repeated three times.

**Figure 7 plants-13-00555-f007:**
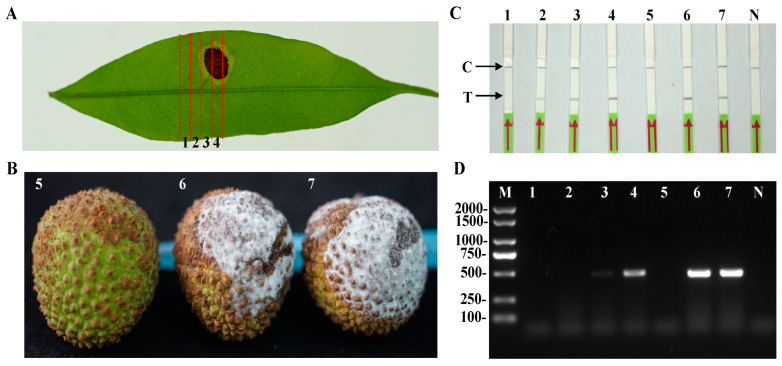
Detection of *Peronophythora litchii* in artificially inoculated lychee tissues using the RPA-LF assay. (**A**) Representative lychee leaf infected by *P. litchii* at 5 days postinoculation (dpi). The infected leaves were divided into four sections (1, 2, 3, and 4) for DNA extraction and detected through the RPA-LF assay; (**B**) representative lychee fruits infected at 5 days postinoculation (dpi); (**C**) detection of *P. litchii* from healthy and infected lychee tissues using the RPA-LF assay; (**D**) conventional PCR detection of *P. litchii* from healthy and infected lychee tissues. 1–4: different regions of infected lychee leaves; 5: non-inoculated control lychee fruits; 6–7: *P. litchii*-inoculated fruits at 5 days postinoculation (dpi). N: negative control (nfH_2_O), M: DL 2000 DNA marker, C: control line, T: test line.

**Table 1 plants-13-00555-t001:** List of *Peronophythora litchii* and other oomycete and fungal species tested for specificity of the RPA-LF assay.

No.	Species	Host	Origin	Number of Isolates	RPA-LFD
1	*Peronophythora litchii*	*Litchi chinensis*	Fujian	2	+
			Hainan	1	+
			Guangdong	3	+
			Guangxi	1	+
2	*Phytophthora citrophthora*	*Citrus reticulata Blanco*	Shanxi	1	-
3	*P. sojae*	*Glycine max*	Fujian	1	-
4	*P. capasici*	*Capsicum frutescent*	Fujian	1	-
5	*P. drechsleri*	*Beta vularis*	Fujian	1	-
6	*P. cinnamomi*	*Cinnamonmum cassia*	Shanxi	1	-
7	*P. infestans*	*Solanum tuberosum*	Fujian	1	-
8	*P. cactorum*	*Malus pumila*	Shanxi	1	-
9	*P. colocasiae*	*Colocasia esculenta*	Fujian	1	-
10	*Pythium aphanidermatum*	*Cucumis sativus*	Fujian	1	-
11	*Colletotrichum gloeosporioides*	*Litchi chinensis*	Fujian	1	-
12	*Colletotrichum truncatum*	*Vigna unguiculata*	Fujian	1	-
13	*Fusarium oxysporum*	*Musa paradisiacal*	Fujian	1	-
14	*Botrytis cinerea*	*Solanum lycopersicum*	Fujian	1	-
15	*Alternaria solani*	*Solanum lycopersicum*	Fujian	1	-
16	*Rhizoctonia solani*	*Oryza glaberrima*	Fujian	1	-
17	*Sclerotinia sclerotiorum*	*Brassica napus*	Fujian	1	-

+: positive amplification; -: negative amplification.

**Table 2 plants-13-00555-t002:** Primers and probe used for the RPA-LF and conventional PCR assay of *Peronophythora litchii*.

Primer Name	Sequences (5′~3′)	Purpose
PlRPALF-F	GAAGACACTGTCAGCACTCTAAACTACTAC	RPA-LF
PlRPALF-R	[Biotin]-GAACCAATCTGGTGAGCCATCGTGACGCAA	
PlRPALF-P	[FAM]-CTACTACCCCATGTACATGGATGTCCACGT-[THF]-ATGATCTACATTGGT-[C3 spacer]	
Yph1F	CGACCATKGGTGTGGACTTT	PCR
Yph2R	ACGTTCTCMCAGGCGTATCT	

## Data Availability

The data that support the findings of this study are openly available within this manuscript.

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
