# Peer review of "Rapid Visual Detection of Peronophythora litchii on Lychees Using Recombinase Polymerase Amplification Combined with Lateral Flow Assay Based on the Unique Target Gene Pl_101565"

_plants, 2024, doi:10.3390/plants13040555_

Round 1

Reviewer 1 Report

Comments and Suggestions for Authors

The RPA-LF technique is very interesting as a suitable technique for fast field screening, but this possible use is not sufficiently emphasised in the MS. This is certainly a strength, certainly not the sensitivity of the method.

Indeed, much attention is paid to the speed, sensitivity and specificity of the method, but the comparison is then made with a traditional PCR instead of a LAMP. Ref. [11] shows, for example, that the sensitivity of a LAMP (10 pg of P. litchii genomic DNA per reaction) is definitely 100 times higher than that of a conventional PCR, so the comparison with a conventional PCR is inappropriate. This also applies to the experiment in Figure 5 where is opportune to keep the presence of lychee genomic DNA constant while reducing the amount of Peronophythora litchii DNA.

Furthermore, in order to facilitate reading and understanding of the work:

(a) Figure 1 must be divided into two new figures otherwise 1 B is illegible;

b) the two parts of Figure 2 should be aligned side by side, indicating what has been added for each column;

c) in Figure 4 the numbers at the top should be replaced with the names of the different oomycetes / fungi;

d) in Figure 5 the numbers should be replaced by indicating what they represent;

e) in the M&Ms it is indicated '....HMMER searches based on pfam00909 were conducted .....' but since pfam00909 is the amonium transporter family it should be explained why the search was already targeted to amonium transporter genes;

f) the conclusions must be revised in relation to the above, and above all a sensitivity of 1 pg of P. litchii gDNA cannot be indicated without indicating a volume in which DNA is present, whereas it must be specified that this result was obtained in the absence of DNA from lychee. Such an evaluation is perfect for a microbiological journal but certainly not for Plants.

Comments on the Quality of English Language

The English is acceptable.

Author Response

Reviewer 1

The RPA-LF technique is very interesting as a suitable technique for fast field screening, but this possible use is not sufficiently emphasised in the MS. This is certainly a strength, certainly not the sensitivity of the method.

Indeed, much attention is paid to the speed, sensitivity and specificity of the method, but the comparison is then made with a traditional PCR instead of a LAMP. Ref. [11] shows, for example, that the sensitivity of a LAMP (10 pg of P. litchii genomic DNA per reaction) is definitely 100 times higher than that of a conventional PCR, so the comparison with a conventional PCR is inappropriate. This also applies to the experiment in Figure 5 where is opportune to keep the presence of lychee genomic DNA constant while reducing the amount of Peronophythora litchii DNA.

Response: We thank the reviewer for pointing out this issue. Compared with traditional PCR, LAMP and other techniques, the most obvious advantage of RPA-LF technique is that the detection results are observed within 5-20 min at 25-43 °C. In this study, the RPA-LF assay developed could be performed effectively at 28 to 46°C within 15 to 25 minutes. In addition, the assay effectively detected P. litchii in infected plant tissue samples. Thus, the RPA-LF assay has the great potential to be developed into a portable detection kit for field applications in detecting P. litchii. As suggested by the reviewer, we have revised the manuscript accordingly.

In the sensitivity evaluation using gDNA, the detection lower limit for the established RPA-LF assay was 1 pg in a 25 μL reaction system. It was 10 times more sensitive than a LAMP assay reported by Kong et al., (2021) [1]. Therefore, the sensitivity of the RPA-LF assay reported here was adequate, whether compared to a conventional PCR or to a LAMP assay.

The sensitivity analyses of numerous RPA-LF assays developed for detecting different pathogens were performed using serial dilutions of pathogen genomic DNA as reaction templates, such as Pseudomonas syringae pv. actinidiae [2], Ustilaginoidea virens[3], Phytophthora sojae [4], Ph. infestans [5], Ph. capsici [6], Ph. parasitica [7], and Pythium [8]. Similarly, the detection limit of LAMP assay was determined using serial dilutions of P. litchii genomic DNA [1]. Therefore, this study evaluated the sensitivity of the RPA-LF assay using various concentrations of P. litchii genomic DNA, consistent with previous research.

  1. Kong, G.; Li, T.; Huang, W.; Li, M.; Shen, W.; Jiang, L.; Hsiang, T.; Jiang, Z.; Xi, P. Detection of Peronophythora litchii on lychee by loop-mediated isothermal amplification assay. Crop Protection 2021, 139.
  2. Yang, Y.; Peng, Q.; Yang, Y.; Zhuang, Q.; Xi, D. Recombinase polymerase amplification-lateral flow (RPA-LF) assay for rapid visual detection of Pseudomonas syringae pv. actinidiae in kiwifruit. Crop Protection 2023, 172.
  3. Jia-cheng, X.; San-lian, W.; Yue, L.; Yuan-di, X.; Jing, Y.; Ting, Z.; Jia-jia, C.; Zheng-guang, Z.; Dan-yu, S.; Hai-feng, Z. Rapid detection of the rice false smut fungus Ustilaginoidea virens by lateral flow strip-based recombinase polymerase amplification assay1. Journal of Integrative Agriculture 2023.
  4. Dai, T.; Yang, X.; Hu, T.; Jiao, B.; Xu, Y.; Zheng, X.; Shen, D. Comparative Evaluation of a Novel Recombinase Polymerase Ampli-fication-Lateral Flow Dipstick (RPA-LFD) Assay, LAMP, Conventional PCR, and Leaf-Disc Baiting Methods for Detection of Phy-tophthora sojae. Front Microbiol 2019, 10, 1884.
  5. Lu, X.; Zheng, Y.; Zhang, F.; Yu, J.; Dai, T.; Wang, R.; Tian, Y.; Xu, H.; Shen, D.; Dou, D. A Rapid, Equipment-Free Method for De-tecting Phytophthora infestans in the Field Using a Lateral Flow Strip-Based Recombinase Polymerase Amplification Assay. Plant Disease 2020, 104, 2774-2778.
  6. Yu, J.; Shen, D.; Dai, T.; Lu, X.; Xu, H.; Dou, D. Rapid and equipment‐free detection of Phytophthora capsici using lateral flow strip‐based recombinase polymerase amplification assay. Letters in applied microbiology 2019, 69, 64-70.
  7. Wang, R.; Zhou, R.; Meng, Y.; Zheng, J.; Lu, W.; Yang, Y.; Yang, J.; Wu, Y.; Shan, W. Specific detection of Phytophthora parasitica by recombinase polymerase amplification (RPA) assays based on a unique multi-copy genomic sequence. Plant Disease 2023.
  8. Jeevalatha, A.; Zumaila, F.; Biju, C.N.; Punya, K.C. Duplex recombinase polymerase amplification assay for simultaneous detection of Pythium spp. and Ralstonia pseudosolanacearum from ginger rhizomes. Crop Protection 2022, 161.

Furthermore, in order to facilitate reading and understanding of the work:

(a) Figure 1 must be divided into two new figures otherwise 1 B is illegible;

Response: As suggested by the reviewer, the Figure 1 has been divided into two new figures as Figure 1 and Figure 2 in the revised manuscript.

  1. b) the two parts of Figure 2 should be aligned side by side, indicating what has been added for each column;

Response: Thanks for your suggestion. First, Figure 2 is now corrected to Figure 3. And we have updated the Figure 3 in the revised manuscript as suggested by reviewer.

  1. c) in Figure 4 the numbers at the top should be replaced with the names of the different oomycetes / fungi;

Response: First, Figure 4 is now corrected to Figure 5. And, the Figure 5 has been revised according to the reviewer’s comment.

  1. d) in Figure 5 the numbers should be replaced by indicating what they represent;

Response: Thanks for your suggestion. First, Figure 5 is now corrected to Figure 6. Serial dilutions of P. litchii gDNA were shown at the top section of image.

  1. e) in the M&Ms it is indicated '....HMMER searches based on pfam00909 were conducted .....' but since pfam00909 is the amonium transporter family it should be explained why the search was already targeted to amonium transporter genes;

Response: Thanks for your suggestion. Modifications were made as required according to the reviewer’s comment in lines 299-306.

All the gene sequences from P. litchii were used as queries to perform homology searches against the genomic sequence of Colletotrichum gloeosporioides by using BLAST with an E-value cutoff of 1E-5. Subsequently, all the retrieved genes were identified and confirmed in NCBI database. Among the unique identified genes, Pl_101565 was selected as the candidate diagnostic target as a single-copy gene. The functional analysis of Pl_101565 was identified by conducting a blast search in the InterPro database (https://www.ebi.ac.uk/interpro/) [37].

  1. f) the conclusions must be revised in relation to the above, and above all a sensitivity of 1 pg of litchii gDNA cannot be indicated without indicating a volume in which DNA is present, whereas it must be specified that this result was obtained in the absence of DNA from lychee. Such an evaluation is perfect for a microbiological journal but certainly not for Plants.

Response: Thanks for reminding. We have modified “The sensitivity evaluation revealed its ability to detect as little as 1 pg of P. litchii gDNA” to “The sensitivity evaluation revealed its ability to detect as little as 1 pg of P. litchii gDNA in a 25 μL reaction system.”

Furthermore, the RPA-LF assay effectively detected P. litchii from artificially infected lychee leaves and fruits, which demonstrated that the RPA-LF method is not influenced by gDNA from lychee. Thus, the RPA-LF assay developed here provide a useful tool for the diagnosis of diseases on lychee. We believe that this manuscript is appropriate for the section "Plant Protection and Biotic Interactions" of Plants.

Reviewer 2 Report

Comments and Suggestions for Authors

The paper submitted to me for review is entitled: “Rapid visual detection of Peronophythora litchii on lychee using recombinase polymerase amplification-lateral flow (RPA-LF) assay based on a newly identified unique target gene.” This is undoubtedly an interesting scientific study that deserves publication. The search for all methods for the early detection of pathogens is commendable and will certainly find practical application. Nevertheless, I believe that the work has some shortcomings that are worthy of attention. First of all, I would like to ask you to change the order of the chapters to a more classical order. Introduction, Materials and Methods, Results, Discussion and Conclusions. If you keep this order, the work will be easier to read. Below you will also find information that I think should be added:

Introduction:

- The publication of specific sources or studies confirming production losses in lychees of 20-30% could support the claim about the impact of the disease on yields.

- Identification of specific studies or sources confirming latent P. litchii infection of lychees and its impact on reduced shelf life could support these claims.

- Detailed information about the primers in the LAMP method (number, sequences) and why they are difficult to design could help to understand the limitations of this method.

- Adding information about the properties of the target gene Pl_101565, such as its function or location, may improve readers' understanding.

Materials and methods

Sections 4.1

- No information on the origin of the 25 isolates, e.g. the place where they were isolated. Accurate information about the isolates is important for the reader to understand their representativeness.

- In general, a culture period of 3-5 days is given for oomycetes and fungi with mycelium. More detailed information on how exactly this time was determined and whether this is the optimal period would be helpful.

Section 4.2

- Information should be provided on the quality of the genome sequences used in the analysis and confirmation that the identification of ammonium transport genes in oomycetes was correct.

- The lack of details on the HMMER analysis parameters and the criteria for the selection of target genes may cast doubt on the effectiveness of this method.

Section 4.3

- More detailed information about the primer and probe sequences, such as sequences and lengths, will help to understand the study design and allow others to replicate the experiments.

Section 4.4

- Missing information on the number of replicates of experiments under RPA-LF conditions may be considered incomplete. Emphasising that the experiments were performed at least in triplicate will increase the reliability of the results.

Section 4.5

- Lack of information on the selection of specific optimisation conditions (temperature and time) may cast doubt on the effectiveness of the method.

Section 4.7

- The lack of precise information on the PCR conditions, such as the number of amplification cycles, may cast doubt on the reliability of the comparison with RPA-LF.

Section 4.8

- The lack of information on the number of lychee samples subjected to the experiment and the specific culture conditions may be important in assessing the practical utility of the method.

Result

Section 2.1

- There is a lack of information about the bioinformatics tools used and the exact criteria for identifying this gene in other oomycetes. Explaining these details will help you understand the identification process and confirm its accuracy.

Section 2.3

- The lack of information on the number of replicates of the experiments when optimising the RPA-LF conditions may cast doubt on the certainty of the results. Adding this information would increase the credibility of the study.

Section 2.5

- Lack of information on the conditions for performing the PCR, such as the number of amplification cycles, may cast doubt on the comparability of the results between RPA-LF and PCR. Precise data on the PCR conditions are important to assess the efficacy of both methods.

Experimental control

- Lack of information on the number of replicates of experiments in different phases of the study, such as optimisation of conditions, specificity, sensitivity, may raise doubts about the repeatability and reliability of the results.

Discussion

- In the "Discussion" section, you should indicate how many times each experiment was repeated to assess the repeatability of the results and the certainty of the data obtained.

- It is worth indicating the specific conditions under which RPA-LF was compared with LAMP and PCR so that readers can judge whether differences in the results are due to different experimental conditions.

- Providing specific reasons for the choice of the Rh-type ammonium transporter gene, Pl_101565, as a marker, including evidence of its uniqueness and diversity among the Oomycetes, will strengthen the argument in favour of the marker used.

- It would be useful to consider the limitations of the method and situations in which it may be less effective. This will allow readers to better understand the conditions under which RPA-LF may be best utilised.

Conclusions

- In the final summary, it is worth reflecting on what concrete benefits the results of this study bring to practise, particularly in the context of agriculture and disease management.

I believe that clarifying these issues will enrich the work and facilitate understanding of the overall context of the research conducted.

Author Response

The paper submitted to me for review is entitled: “Rapid visual detection of Peronophythora litchii on lychee using recombinase polymerase amplification-lateral flow (RPA-LF) assay based on a newly identified unique target gene.” This is undoubtedly an interesting scientific study that deserves publication. The search for all methods for the early detection of pathogens is commendable and will certainly find practical application. Nevertheless, I believe that the work has some shortcomings that are worthy of attention. First of all, I would like to ask you to change the order of the chapters to a more classical order. Introduction, Materials and Methods, Results, Discussion and Conclusions. If you keep this order, the work will be easier to read. Below you will also find information that I think should be added:

Response: Thank you for your positive comments and valuable suggestions to improve the quality of our manuscript! First, we need clarify that the order of the chapters is the fixed format for the journal of Plants. Moreover, we have modified the concerned sections as suggested by the reviewer in the revised manuscript, and all the changes are marked in red color.

Introduction:

- The publication of specific sources or studies confirming production losses in lychees of 20-30% could support the claim about the impact of the disease on yields.

Response: Thanks for reminding. A reference was added in line 35.

- Identification of specific studies or sources confirming latent P. litchii infection of lychees and its impact on reduced shelf life could support these claims.

Response: Relevant reference has been added in line 39.

- Detailed information about the primers in the LAMP method (number, sequences) and why they are difficult to design could help to understand the limitations of this method.

Response: As suggested by the reviewer, the detailed information about the primers in the LAMP method as follows were added in the revised manuscript. (Lines 52-53)

“However, this method requires the design of four specific primers for six distinct regions on the target gene[13].”

- Adding information about the properties of the target gene Pl_101565, such as its function or location, may improve readers' understanding.

Response: Many thanks for the suggestion. the information about the newly identified gene Pl_101565 as follows was added in the revised manuscript. (Lines 74-77)

In this investigation, the fusion of RPA and LF smethodologies was employed to propose a rapid detection assay for P. litchii, based on the newly identified gene Pl_101565 encoding a Rh-type ammonium transporter, derived from genomic sequence data.

Materials and methods

Sections 4.1

- No information on the origin of the 25 isolates, e.g. the place where they were isolated. Accurate information about the isolates is important for the reader to understand their representativeness.

Response: The related information has been listed in the Table 1.

- In general, a culture period of 3-5 days is given for oomycetes and fungi with mycelium. More detailed information on how exactly this time was determined and whether this is the optimal period would be helpful.

Response: The detailed information was added as follow in line 279.

For DNA extraction, mycelia obtained from each isolate were cultured in 10% liquid V8 medium and potato dextrose broth medium, respectively, at temperatures ranging from 18 to 25°C for 3-5 days until the mycelium covered the Petri dish.

Section 4.2

- Information should be provided on the quality of the genome sequences used in the analysis and confirmation that the identification of ammonium transport genes in oomycetes was correct.

Response: Many thanks for the suggestion. Related information as follows were supplemented in line 292 and 310, respectively.

The published genome sequences of P. litchii (Accession No. GCA_002812785.1) and nine other sequenced pathogens, including Ph. sojae (GCA_000149755.2), Ph. capsici (GCA_030324255.1), Ph. ramorum (GCA_020800215.1), Ph. parasitica (GCA_000247585.2), Ph. cinnamomi (GCA_018691715.1), Ph. infestans (GCA_000142945.1), Ph. palmivora (GCA_008079305.1), Hyaloperonospora parasitica (GCA_029452305.1), Pythium aphanidermatum (GCA_000387445.2), and Colletotrichum gloeosporioides (GCF_011800055.1)were retrieved from the National Center for Bio-technology Information (NCBI) database with the genome coverage ranging from 8.0 to 340.0X [4,32].

The candidate sequences were the confirmed iteratively by comparison the NCBI database.

- The lack of details on the HMMER analysis parameters and the criteria for the selection of target genes may cast doubt on the effectiveness of this method.

Response: Thanks for reminding. The details were supplemented in lines 308-310.

To identify Rh-type ammonium transporter genes, HMMER searches based on pfam00909 were performed within the genomes of nine other oomycete species using a cutoff of 1E-5 [28].

Section 4.3

- More detailed information about the primer and probe sequences, such as sequences and lengths, will help to understand the study design and allow others to replicate the experiments.

Response: The information about the primer and probe sequences were added as follows in lines 314-316.

The forward and reverse primer lengths of RPA primers were both 30-nt, and the designed probe was 46-nt long.

Section 4.4

- Missing information on the number of replicates of experiments under RPA-LF conditions may be considered incomplete. Emphasising that the experiments were performed at least in triplicate will increase the reliability of the results.

Response: Thanks for reminding. It was added in line 338.

Section 4.5

- Lack of information on the selection of specific optimisation conditions (temperature and time) may cast doubt on the effectiveness of the method.

Response: It was added as follows in lines 342-343.

Typically, the RPA reaction can be performed within 20 min at temperatures be-tween 22 and 45°C.

Section 4.7

- The lack of precise information on the PCR conditions, such as the number of amplification cycles, may cast doubt on the reliability of the comparison with RPA-LF.

Response: It was added in lines 368-370.

Section 4.8

- The lack of information on the number of lychee samples subjected to the experiment and the specific culture conditions may be important in assessing the practical utility of the method.

Response: The related information has been added in lines 376-378 and 381-382, respectively.

Result

Section 2.1

- There is a lack of information about the bioinformatics tools used and the exact criteria for identifying this gene in other oomycetes. Explaining these details will help you understand the identification process and confirm its accuracy.

Response: Thanks for reminding. Relevant detailed information has been added as follows in Section 4.3 of Materials and Methods.

To identify Rh-type ammonium transporter genes, HMMER searches based on pfam00909 were performed within the genomes of nine other oomycete species using a cutoff of 1E-5. The candidate sequences were the confirmed iteratively by com-parison the NCBI database.

Section 2.3

- The lack of information on the number of replicates of the experiments when optimising the RPA-LF conditions may cast doubt on the certainty of the results. Adding this information would increase the credibility of the study.

Response: Thanks for reminding. It was added in line 134.

Section 2.5

- Lack of information on the conditions for performing the PCR, such as the number of amplification cycles, may cast doubt on the comparability of the results between RPA-LF and PCR. Precise data on the PCR conditions are important to assess the efficacy of both methods.

Response: Many thanks for the suggestion. The information on the conditions for performing the PCR has been added in lines 368-370. In addition, modifications were made as required in Section 2.5 as follows in lines 162-166.

The results for the sensitivity assay were consistent across all three replicates. Notably, the detection sensitivity of conventional PCR assay was 1 ng of DNA in a 25 μL reaction mixture (Figure 6B). The RPA-LF assay is 1000 times more sensitive than conventional PCR under the conditions as described above, making it highly effective for detecting P. litchii gDNA.

Experimental control

- Lack of information on the number of replicates of experiments in different phases of the study, such as optimisation of conditions, specificity, sensitivity, may raise doubts about the repeatability and reliability of the results.

Response: Many thanks for the suggestion. The information on replicates of experiments in different phases of the study was added in line 134, 146, 162, and 188, respectively.

Discussion

- In the "Discussion" section, you should indicate how many times each experiment was repeated to assess the repeatability of the results and the certainty of the data obtained.

Response: As suggested by the reviewer, the information on replicates of experiments in different phases of this study has been added in the “Results” and “Materials and Methods” section, which is sufficient to assess the repeatability of the results and the certainty of the data obtained.

- It is worth indicating the specific conditions under which RPA-LF was compared with LAMP and PCR so that readers can judge whether differences in the results are due to different experimental conditions.

Response: Many thanks for the suggestion. We have supplemented the specific conditions under which RPA-LF was compared with LAMP and PCR in lines 208-212, 239-242, and 254-256.

- Providing specific reasons for the choice of the Rh-type ammonium transporter gene, Pl_101565, as a marker, including evidence of its uniqueness and diversity among the Oomycetes, will strengthen the argument in favour of the marker used.

Response: Thanks for your suggestion. The related information was added in lines 227-233 as follows.

In this study, the Rh-type ammonium transporter gene, Pl_101565, was identified from the P. litchii genomic sequence. No homologs of gene Pl_101565 were identified in other reference plant and fungal species, suggesting that this gene is specific to oomycete species. Additionally, a comparative analysis of P. litchii and 9 other oomycetes revealed sequence similarities between 44.93% and 79.87%, indicating that gene Pl_101565 exhibited sufficient sequence variability among different oomycetes.

- It would be useful to consider the limitations of the method and situations in which it may be less effective. This will allow readers to better understand the conditions under which RPA-LF may be best utilised.

Response: Thanks for your suggestion. As suggested by the reviewer, the discussion as follows are added in the revised manuscript. (lines 256-260)

However, there are some limitations to RPA assay [18]. One major limitation to con-sider is that the RPA kits are currently only sold by one company, which may have an impact on pricing. Additionally, there is currently no software available for designing primers specifically for RPA, which can lead to time-consuming for primer sequences optimization.

Conclusions

- In the final summary, it is worth reflecting on what concrete benefits the results of this study bring to practise, particularly in the context of agriculture and disease management.

Response: Many thanks for the suggestion. In conclusions, “This innovative method offered a fast and convenient molecular tool for early detection of P. litchii, which could facilitate the timely development of proper management strategies for downy blight control, particularly important for postharvest litchi fruit.” was added in the revised manuscript. (lines 402-405)

I believe that clarifying these issues will enrich the work and facilitate understanding of the overall context of the research conducted.

Response: We gratefully appreciate you for reading our article carefully and giving the above positive comments.

Round 2

Reviewer 1 Report

Comments and Suggestions for Authors

The MS has been revised and much improved, but two points remain to be corrected:

a) the images relating to conventional PCR electrophoresis should be removed from Figures 6 and 7, and at the authors' discretion, also from Figure 3 where, eventually, the image should be described as a simple example. In fact, if the current figures are retained, the reader immediately concludes that the aim of the work is to demonstrate that the RPA-LF assay performs better than a conventional PCR.

b) Improve the quality of the figures and utilise the entire side space of the format (particularly for Figure 2).

Author Response

Reviewer 1

The MS has been revised and much improved, but two points remain to be corrected:

Response: We agree with you and have incorporated the suggestions throughout our paper. Thank you again for your positive comments on our article.

  1. a) the images relating to conventional PCR electrophoresis should be removed from Figures 6 and 7, and at the authors' discretion, also from Figure 3 where, eventually, the image should be described as a simple example. In fact, if the current figures are retained, the reader immediately concludes that the aim of the work is to demonstrate that the RPA-LF assay performs better than a conventional PCR.

Response: Many thanks for the suggestion. As suggested by the reviewer, the image relating to conventional PCR electrophoresis in Figure 6 has been removed in the revised manuscript. However, it is suitable for that the electrophoretic image of the conventional PCR assay in Figure 7, because conventional PCR was used to further confirm the RPA-LF results. Similarly, the image electrophoretic from Figure 3 showed the specific amplification band of the RPA reaction, further confirming the visualize RPA amplicons of lateral flow (LF).

The results, Figure legends, and Materials and Methods were revised accordingly.

  1. b) Improve the quality of the figures and utilise the entire side space of the format (particularly for Figure 2).

Response: As suggested by the reviewer, we have improved the quality of the figures (300 ppi) in the revised manuscript, including Figure 1-7.

Reviewer 2 Report

Comments and Suggestions for Authors

The authors have responded to all my comments and provided reliable and comprehensive answers. In my opinion, the article is suitable for publication in its present form.

Author Response

The authors have responded to all my comments and provided reliable and comprehensive answers. In my opinion, the article is suitable for publication in its present form.

Response: We gratefully appreciate you for reading our article carefully and giving the above positive comments.